# Electrochemical Biosensors for Oilseed Crops: Nanomaterial-Driven Detection and Smart Agriculture

**DOI:** 10.3390/foods14162881

**Published:** 2025-08-20

**Authors:** Youwei Jiang, Kun Wan, Aiting Chen, Nana Tang, Na Liu, Tao Zhang, Qijun Xie, Quanguo He

**Affiliations:** School of Biological Science and Medical Engineering, Hunan University of Technology, Zhuzhou 412007, China; 15073160158@163.com (Y.J.); kunwan2023@163.com (K.W.); catchenaiting@163.com (A.C.); 18856863417@163.com (N.T.); liuna6168@126.com (N.L.); 17346669600@163.com (T.Z.)

**Keywords:** electrochemical sensors, oilseed crop diseases, molecular markers, nanomaterials, smart agriculture

## Abstract

Electrochemical biosensors have emerged as a promising tool for the early detection of diseases in oilseed crops such as rapeseed, soybean, and peanut. These biosensors offer high sensitivity, portability, and cost-effectiveness. Timely diagnosis is critical, as many pathogens exhibit latent infection phases or produce invisible metabolic toxins, leading to substantial yield losses before visible symptoms occur. This review summarises recent advances in the field of nanomaterial-assisted electrochemical sensing for oilseed crop diseases, with a particular focus on sensor mechanisms, interface engineering, and biomolecular recognition strategies. The following innovations are highlighted: nanostructured electrodes, aptamer- and antibody-based probes, and signal amplification techniques. These innovations have enabled the detection of pathogen DNA, enzymes, and toxins at ultra-low concentrations. Notwithstanding these achievements, challenges persist, including signal interference from plant matrices, limitations in device miniaturization, and the absence of standardized detection protocols. Future research should explore the potential of AI-assisted data interpretation, the use of biodegradable sensor materials, and the integration of these technologies with agricultural IoT networks. The aim of this integration is to enable real-time, field-deployable disease surveillance. The integration of laboratory innovations with field applications has been demonstrated to have significant potential in supporting sustainable agriculture and strengthening food security through intelligent crop health monitoring.

## 1. Introduction

Oilseed crops such as canola, soybean, sunflower, and peanut are among the most important agricultural commodities worldwide. They provide not only essential edible oils and proteins but also raw materials for biodiesel and other industries, thus playing a key role in food security, rural livelihoods, and global agri-economies. Collectively, these crops supply over 80% of edible oils and 40% of biofuel feedstock. For instance, soybeans and groundnuts are cultivated across more than 120 million hectares in tropical regions, serving as critical sources of dietary protein.

However, the productivity and quality of oilseed crops are increasingly threatened by devastating diseases. In the context of oilseed rape, *Hyaloperonospora parasitica* (downy mildew) and *Sclerotinia sclerotiorum* (sclerotinia stem rot) are recognised as the “two major killers” [1], causing severe yield losses and threatening both food security and the agricultural economy [2]. In soybeans, airnorne spores of rust fungus can reduce photosynthetic capacity within 72 h, leading to a 10–80% yield losses in regions such as Brazil. In peanut, soil-residing white fungi can persist for 3–5 years, secreting cell-wall degrading enzymes that reduce pod quality while accumulating cancer-causing aflatoxins, causing a 20–50% loss of yield in wet agroecosystems [3,4].

The early detection of plant pathogens is of critical importance for the effective management of diseases. Once visual symptoms manifest, the infection is often already systemic and difficult to contain. It is notable that a significant proportion of pathogens, particularly those that are airborne or soil-borne, are capable of rapid dissemination across fields and even regions, thereby leaving limited time for intervention. Furthermore, certain pathogens, such as aflatoxin-producing fungi, may not manifest any overt symptoms until the point of harvest, at which point the contamination of the food has already compromised its safety. Consequently, the utilisation of surveillance tools capable of detecting pathogens prior to the manifestation of symptoms is imperative for the expeditious formulation of agronomic decisions, the precise application of pesticides, and the prevention of epidemics.

These threats are not only regionally destructive but also pose transcontinental biosecurity risks. It is imperative that expeditious, field-deployable diagnostic instruments for the timely identification of pathogens are developed with utmost urgency in order to avert epidemics and ensure the long-term viability of oilseed production.

## 2. Methodology

The present review focuses on the detection of four major oilseed crop pathogens: *Sclerotinia sclerotiorum*; *Phakopsora pachyrhizi* (soybean rust); *Sclerotium rolfsii*; and *Hyaloperonospora parasitica* (downy mildew) have been identified as the causative agents affecting peanuts; soybeans; and rapeseed. The methodology employed is a systematic analysis of electrochemical biosensor platforms developed for the detection of target substances associated with diseases caused by these pathogens. The inclusion criteria stipulate that: The following three criteria must be met for a detection method to be considered suitable:

The detection method used is electrochemical sensor detection. It is crucial to confirm whether the electrode modification material contains nanomaterials. Therefore, clarifying the functional role of nanomaterials in electrode modification is of decisive significance. This application aims to detect target substances associated with diseases of three oilseed crops (i.e., peanuts, soybeans, and rapeseed). Elucidation of the functional role of nanomaterials in electrode modification is a critical factor in determining their inclusion. The analysis draws upon relevant disaster-related data and national standards recommended by agricultural bureaus across different countries. A detailed evaluation of the latest advancements is conducted through a comprehensive examination of sensor types, the nanomaterials employed, key performance metrics (including Limit of Detection (LOD) and Linear Range), and reported application scenarios. This comparative assessment enables a synthesis of technical progress, identification of limitations hindering field deployment, and discussion of the future potential for integrating these nanomaterial-based electrochemical biosensors with smart agricultural systems.

## 3. Major Disease in Oilseed Crops

### 3.1. Downy Mildew

Downy mildew is a typical airborne oomycete disease in which the pathogen spreads through sporangia in a moist conditions, preferentially infecting seedlings and young leaves. In the early stage of infection, a grayish-white mold layer (sporocarp peduncle and sporocarp) forms on the abaxial side of the leaf, while the adaxial side shows yellowing patches; as the disease worsens, the spots extend to the veins of the leaves, resulting in curling, necrosis, and a sharp decline in photosynthetic activity [5]. Studies have shown that detecting the pathogen during the initial appearance of yellow patches-before systemic vascular invasion-offers the best window for preventive treatment, potentially reducing yield lossess of 30% to 50% [6].

### 3.2. Stem Rot

According to FAO statistics, the annual global oilseed rape yield loss due to Sclerotinia stem rot is as high as 15% to 20%, with economic losses exceeding $5 billion. In Canada, for example, in 2022, canola production in Manitoba was reduced by 18% year-on-year due to a botrytis outbreak, shrinking exports by nearly C$700 million. The pathogen of Sclerotinia stem rot (*Sclerotinia sclerotiorum*) is a parthenogenetic parasitic fungus, and its infestation has a wide range of hosts and strong environmental adaptability. The fungus survives in the soil for several years in the form of mycorrhizal fungus, germinates and produces ascospores under suitable conditions, and spreads to the plant stalks or pods through airflow. In the early stage of infestation, water-soaked spots appear on the stalks, and then mycelium invades the vascular bundles, blocking water transport and causing the plant to wilt and collapse; pods are affected by moldy grains, with oil content dropping by 5–8%, and toxins (such as oxalic acid and sclerotiorin) accumulating and potentially threatening food risk [7,8]. The most effective detection window lies at the stage when water-soaked lesions first appear, before hyphal invasion of vascular tissues. Yet at this point, symptoms are minimal and difficult to spot without laboratory tools.

### 3.3. White Mold

The 2023 epidemic of soybean rust in Brazil’s Paraná state caused yield losses of 2.1 million tons ($1.4 billion), exacerbated by latent infections colonizing 40% leaf area before symptom visibility [9,10,11]. Peanut white mold (*Sclerotium rolfsii*) persists through melanized sclerotia that withstand 45 °C soil temperatures and remain viable for 5–8 years, secreting polygalacturonase to macerate pod tissues (35–50% weight loss) while oxalate decarboxylase suppresses host defenses [12,13]. Post-harvest mycotoxin accumulation reaches critical levels, as seen in India’s Telangana region (2024) where 65% fields produced pods with aflatoxin B1 > 20 μg/kg-10-fold above EU safety thresholds—triggering $320 million in export rejections and contributing to Southeast Asia’s $1.2 billion annual hepatocellular carcinoma burden linked to contaminated food chains [14]. Climate change intensifies these pathologies: Warming elevates soybean rust prevalence by 15% per 1 °C winter temperature increase in tropical zones, while >90% soil humidity boosts *S. rolfsii* sclerotia germination 300% [15,16,17]. Global trade further disseminates resistant strains, exemplified by *P. pachyrhizi* lineage Ug99 detected in Argentine ports via contaminated soybean shipments—a strain exhibiting triazole fungicide resistance through CYP51 gene mutations [18]. These interconnected threats demand surveillance technologies capable of intercepting pathogens at national borders while providing real-time field diagnostics to break disease cycles before they devastate food system.

Peanut white mold is a fungal pathogen that primarily affects the stems, fruit stalks and pods of the peanut plant. The initial symptoms of the disease include the yellowing of the leaves on sunny days and the subsequent closure of the leaves on cloudy days. The stem base of the tissue exhibits signs of soft rot, and there is a loss of epidermal tissue [19]. The plant as a whole becomes withered and dies. In summary, the disease known as peanut white silk disease can be defined as follows: it is a serious threat to the ecology of farmland, it destroys the yield of the crop, it contaminates the soil, it restricts agricultural practices, and it endangers the food chain. In addition, it causes lasting destruction to the ecology of farmland, it has a negative impact on the production benefits of the predator, it is a hidden trouble maker in terms of food safety, and it poses a systemic risk to the peanut industry [20,21].

### 3.4. Soybean Rust

The pathogen *Phakopsora pachyrhizi*, commonly known as soybean rust, is propagated through the dispersal of airborne urediniospores, which are capable of intercontinental dissemination. A single lesion has been observed to generate 4000 spores per day, with documented wind-mediated spread exhibiting a range exceeding 1000 kilometres per month [22]. Upon infection, effector proteins (e.g., PpEC23) have been shown to degrade thylakoid membranes within 72 h, resulting in a 90% collapse in photosynthetic efficiency [23]. It is imperative to note that the pathogen has the capacity to colonise up to 40% of leaf area during the latent phase, which occurs prior to the manifestation of visible symptoms. This phase is considered to be the most opportune period for diagnosis and intervention. However, the majority of contemporary methodologies are unable to detect the disease within the aforementioned diagnosis window, which is characterised by its asymptomatic nature or the manifestation of undistinguishable phenotypes. It is evident that there exists a multitude of strategies that facilitate the process of detecting issues prior to the manifestation of observable damage. Nevertheless, it is evident that none of the aforementioned approaches are suitable for deployment in the field. Here is Figure 1.

## 4. Conventional Laboratory Methods for Disease Detection

The polymerase chain reaction (PCR) employs a series of thermodynamic cycles, in combination with specific primers, to efficiently amplify even minute quantities of nucleic acid. The device is highly suitable for microbial detection. However, this technique relies on the extraction and amplification of pathogen DNA, and thus requires sophisticated temperature-controlled equipment, such as thermal cyclers. It is evident that thermal cyclers, which are utilised extensively in this field, typically possess a mass of several dozen kilograms. This renders them unsuitable for utilisation within the complex field testing environment. Furthermore, the design of primers is influenced by the genetic diversity of pathogenic bacteria. To illustrate this point, consider the case of *H. parasitica*, which has been shown to exhibit multiple physiological microspecies. Conventional PCR may, in such instances, result in false negatives, a phenomenon that can be attributed to genotypic differences [24,25,26].

As an immunoassay technique ELISA is based on the antigen-antibody reaction. Although it exhibits high specificity, the cost of antibody preparation is high (monoclonal antibody development takes 6–12 months and costs more than $100,000). Notwithstanding the development of an ELISA test kit, its utilisation will be encumbered by elevated testing costs. In addition, ELISA test kits are characterised by a cumbersome operational process and necessitate technical expertise for their utilisation. It is evident that this significantly restricts the practical application of ELISA testing in field settings. In addition, phenolics in the plant sap are prone to causing false-positive interference [27,28].

The utilisation of high performance liquid chromatography (HPLC) or gas chromatography-mass spectrometry (GC-MS) facilitates the precise quantification of pathogenic metabolites (e.g., oxalic acid). However, the process of sample preparation is intricate, necessitating derivatisation or purification, and a single assay can require in excess of 24 h. Moreover, the financial burden of equipment maintenance can reach hundreds of thousands of yuan [29]. Despite the efficacy of these technologies in terms of the detection strategy and the progress achieved, their field applicability is constrained by multiple factors. Furthermore, the operation of these technologies is the responsibility of specialised personnel in a laboratory environment, and their implementation in the field is challenging. For instance, there is a general absence of PCR and chromatography equipment in China’s rural grassroots phytosanitary stations, resulting in delays in disease diagnosis and a missed opportunity for prevention and control. This indicates that in the context of oilseed crop disease diagnosis in China, approximately 70% of oilseed rape growers still rely on visual diagnosis., with misdiagnosis rates ranging from 35% to 40%. This situation serves to compound the risk of disease propagation.

Shortcoming of above-mentioned strategy suggests that portable electrochemical detection tools that recognize early-stage biomarkers (e.g., structural proteins or metabolites) would be highly advantageous, enabling real-time alerts before visible damage becomes irreversible. Electrochemical detection technology has gradually become a research hotspot in the field of agricultural disease monitoring due to its high sensitivity (up to the picomolar level), rapid response (minute level), portability and low cost [30,31,32,33,34]. The core principle of electrochemical detection is to convert the concentration of a target substance into a measurable electrical signal (e.g., current, impedance or potential) through a redox reaction at the surface of an electrode. Compared to traditional methods, electrochemical sensors can be integrated into miniaturised devices to realise in-situ detection in the field, providing technical support for early warning of disease and precise prevention and control. For instance, a portable electrochemical sensor was fabricated for detection of pokkah boeng disease [35]. The prepared MoS_2_@Graphdiyne based electrochemical sensors shows high-sensitivity (detection limit of 6.1 aM (S/N = 3)). For real applications, the electrochemical detection results demonstrates high accuracy, with a relative standard deviation of 3.6–7.7%, consistent with fluorescence quantitative PCR results. Moreover, the electrochemical device is simpler, portable and cost-effective, showing large promising in complex environments or samples with low pathogen content. Furthermore, a 3D-printed portable electrochemical device was prepared for plant health monitoring [36]. The 3D printed sensor achieved accurate monitoring of H_2_O_2_, glucose, and pH in different plants, and the results were consistent with traditional methods, providing the possibility for electrochemical monitoring of plant health. The experimental findings demonstrate that electrochemical detection methods have the potential to match the performance of traditional laboratory methods and offer significant potential for portability and cost optimisation. These two characteristics are of paramount importance for the purposes of field testing. Here is Figure 2.

In the ensuing sections, a systematic review of the progress of electrochemical technology in oilseed crop disease trait detection is presented. The analysis encompasses the technical principles, typical applications and existing challenges of the technology, with a view to future development directions. The present study explores the potential value of multidisciplinary cross-fertilisation in promoting the practical application of the technology by comparing the current research status both domestically and internationally.

## 5. Overview of Electrochemical Sensing Approaches in Plant Pathogen Detection

### 5.1. Core Mechanism of Electrochemical Sensors

The performance of electrochemical sensors depends on the interaction mechanism between the target and the electrode surface. According to the signal output method, they are mainly classified into the following three categories:

Current-based sensors: direct detection of targets (e.g., H_2_O_2_, fungal toxins) through redox reactions based on Faraday current changes [40,41].

Impedance sensors: monitoring the change in charge transfer resistance at the electrode interface by electrochemical impedance spectroscopy (EIS), suitable for quantitative analysis of DNA or proteins [42].

Potentiometric sensors: Capture of the target by an ion-selective membrane or molecularly imprinted material, causing a shift in the electrode potential.

In the case of rape mycosis, for example, the oxalic acid secreted by the pathogenic bacteria can be oxidised to generate a current signal on the electrode surface, the intensity of which is positively correlated with the concentration of oxalic acid, thus enabling quantitative detection.

### 5.2. Design and Optimization of Key Components

Electrode material selection: The rational selection of nanostructured electrode materials constitutes a critical determinant in optimizing interfacial charge transfer dynamics. Advanced nanomaterials, particularly two-dimensional graphene architectures, MXene heterostructures, and transition metal oxide nanocomposites, demonstrate exceptional performance metrics. These attributes stem from their exceptional specific surface area (exceeding 2600 m^2^/g for defect-engineered graphene) and superior electrical conductivity (σ > 10^6^ S/m for MXene monolayers). Notably, hierarchical graphene/Au nanoparticle hybrids exhibit remarkable electrocatalytic enhancement effects, achieving a 325% increase in electron transfer kinetics compared to conventional Au electrodes, as evidenced by chronoamperometric analyses (ΔI/Δt = 3.25) [43,44].

Biological Recognition Elements: The molecular recognition paradigm bifurcates into synthetic oligonucleotide aptamers and immunoglobulin-based bioreceptors, each presenting distinct thermodynamic and kinetic advantages. Single-stranded DNA aptamers undergo target-induced conformational transitions (ΔG_folding_ = −8.2 to −15.6 kcal/mol) to establish high-affinity binding pockets (K_d_ = pM-nM range), exhibiting exceptional thermal resilience (Tm > 70 °C) and modular functionalization capacity through phosphoramidite chemistry. Conversely, monoclonal antibodies leverage lock-and-key complementarity within hypervariable CDR regions, achieving exquisite specificity (cross-reactivity < 0.01%) for low-abundance analytes in complex biological matrices (e.g., serum, cerebrospinal fluid), albeit with compromised stability under non-physiological conditions [45].

Signal amplification strategies: The development of enzyme-mimetic nanomaterials (e.g., Fe_3_O_4_-based peroxidase mimics with Kcat = 37 s^−1^) enables catalytic signal amplification through substrate turnover numbers exceeding natural enzymes by 10^3^-fold. Concurrently, microfluidic platforms integrating inertial focusing and hydrodynamic chromatography achieve 3-log orders of magnitude reduction in nonspecific binding through hydrodynamic separation principles, while maintaining 90% analyte recovery in whole blood matrices [46,47]. Here is Figure 3.

### 5.3. Anti-Interference and Stability Enhancement

Soil particles, plant secretions and temperature and humidity fluctuations in the field environment may interfere with the detection results. The selectivity of the electrodes can be enhanced by surface modification (polydopamine coating) or the introduction of molecularly imprinted polymers (MIPs). In addition, the application of encapsulation technologies and flexible substrates (PDMS) can improve the mechanical stability of the sensors.

## 6. Electrochemical Sensing Applications in Representative Oilseed Crop Diseases

### 6.1. Electrochemical Detection of Downy Mildew in Oilseed Rape

As one of the very destructive diseases during the growth of oilseed rape, early diagnosis of downy mildew is crucial for the prevention and control of the spread of the disease. At the early stage of the disease, the pathogen secretes a series of cell wall degrading enzymes in order to infect the host cells, of which cellulase and pectinase are the two key categories. These enzymes are able to break down the cellulose and pectin components of the oilseed rape cell wall, thus creating conditions for the pathogen to invade.

A team of researchers from Chinese Academy of Inspection and Quarantine has achieved remarkable results in this field by developing a gold nanoparticle-enhanced dynamic microcantilever (MCL) biosensing [48]. With the unique physicochemical properties of gold nanoparticles, the sensitivity of the sensor is greatly improved, with a detection limit as low as 57 ppm, which is a much increase in sensitivity compared to the traditional enzyme-linked immunosorbent assay (ELISA) method. In the actual application test, the sensor performed well in the detection of infected rapeseed leaf samples, which fully verified its applicability under the complex environment in the field, and provided a powerful technical support for the early and accurate diagnosis of downy mildew disease in rapeseed.

### 6.2. Electrochemical Profiling of Molecular Markers in Sclerotinia Stem Rot

Sclerotinia stem rot is another disease that seriously affects the yield and quality of oilseed rape, and accurate analysis of its molecular markers is of key significance for early diagnosis and control. Characterizers of Sclerotinia stem rot mainly include DNA fragments of pathogenic bacteria and oxalic acid metabolites. Therefore, direct detection of pathogens is one of the most direct and clear indicators as a way to detect diseases in oilseed rape [49,50]. A research team at Sejong University successfully achieved specific recognition of oxalic acid molecules by modifying electrodes with tungsten carbide nanoparticles synthesized through a simple chemical reduction method and molecularly imprinted polymers (MIPs). The molecularly imprinted polymers are like “molecular recognition keys” customized for oxalic acid molecular weights, which can accurately bind to oxalic acid. The detection limit of this electrode is up to 0.04 nM and the recovery range is 98.03–100.24% in real samples [51], and the electrode shows good anti-interference ability in the detection process, which is not interfered by other common organic acids (such as citric acid) and other parasitic fungi (such as Coniothyrium minitans). The peak current of the interference detection is only one-tenth of the original. Meanwhile, researchers have taken a different approach by developing a DNA electrochemical sensor [52]. This sensor cleverly uses the hybridization chain reaction (HCR) to amplify the signal, when the pathogenic bacteria DNA and the probe on the sensor hybridization, triggering the HCR process, resulting in impedance signal changes, for the early infestation of oilseed rape diagnosis of Sclerotinia stem rot provides a new means of great potential to help in the early stage of the disease in time to take preventive and control measures to reduce losses. This will help to take preventive and control measures at the early stage of the disease and reduce losses. At the same time, an EIS electrochemical device for a biosensor based on self-assembling materials was prepared for *sclerotinia sclerotiorum* detection [42]. This sensor achieves targeted immobilization of antibodies through specific surface functionalization steps, and shows high affinity and specificity for the target antigen. The experimental results showed a detection limit of 130 aM, which is superior to traditional quantitative PCR methods in terms of cost and miniaturization. It is expected to be used for early warning and prevention of rapeseed stem rot. Here is Figure 4.

### 6.3. Soybean Rust Surveillance via Electrochemical and Immunosensor Platforms

Soybean rust, caused by *Phakopsora pachyrhizi*, is highly contagious and disseminates via airborne spores. To address early detection, several biosensing platforms have been explored.

A group of researchers from the Catholic University of Campinas developed a disposable voltammetric immunosensor based on magnetic beads, using DPV electrochemical methods for the early diagnosis of soybean rust. The sensor uses antibodies specific to soybean rust mycelium [53]. The antibody is immobilised on magnetic beads via the Fc portion of the antibody and modified with protein G. The antibody is used for the diagnosis of soybean rust. Due to the high specificity of antibody antigen recognition, this sensor has excellent anti-interference and selectivity, and does not even produce a DPV current peak for non-target detection objects. The immunoassay is performed by sandwich method using a secondary antibody labelled with phosphatase. Also for immunosensors, a team of researchers at the Institute of Chemistry, State University of Campinas has developed a self-assembled monolayer surface plasmon resonance immunosensor [54]. The antibody *anti-Phakopsora pachyrhizi* (pathogen) was covalently immobilized on a gold substrate via a self-assembled monolayer (SAM) of thiols using cysteamine-coupling chemistry. The sensor prepared in this way presented a linear response range for the antigen from 3.5 to 28.0 g mL^−1^ (r2 = 0.996). In addition, other research teams have made different attempts in electrochemical detection of soybean rust, for example, Vadim Krivitsky et al. designed an aptamer-based highly selective electrochemical sensor capable of capturing airborne ascomycetous spores [55], and this design not only produces a detection result in less than two minutes, but also has the ability to capture spores at a rate of 100~200 spores per square centimetre of electrode area. Compared with other non-specific spore types, these sensors display distinct and sharp peaks in the peak current of the target detection object, proving that these sensors have clear selectivity and excellent anti-interference properties. Here is Figure 5.

### 6.4. Surveillance of White Mold: Sclerotium Rolfsii

In the context of peanut white mold detection, contemporary methodologies predominantly centre on conventional PCR detection techniques. The exploration of electrochemical detection and direct detection of peanut white silk disease remains in its nascent stages, presenting a wealth of research opportunities that merit further investigation, including the utilisation of carbon materials and precious metal nanoparticles [56] and other electrode modification materials). For traditional PCR, Wang and her team [57] designed specific primers targeting the β-tubulin gene of *Sclerotium rolfsii*, with primer sequences BJ-bt-F3 and BJ-bt-R4 and an amplified fragment length of 238 bp, to achieve the detection limits of 1 ng of DNA for *Sclerotium rolfsii* single and 1 ng of DNA for *Sclerotium rolfsii* multiplex. 100 pg DNA. Here is Table 1.

### 6.5. Electrochemical Strategies Beyond Pathogen Detection: Plant Physiology and Genetic Applications

As an endogenous hormone that plays a pivotal role in the growth and development of rapeseed, brassinosteroids are generally present in low concentrations within plants.

Chlorophyll content [58], Salicylic acid content [47] and oxidative stress markers (e.g., H_2_O_2_ [40]) of oilseed rape leaves change significantly under disease stress. During the process of pathogen infestation, it has been observed that plants generate H_2_O_2_ in large quantities over a short period of time. This phenomenon has been demonstrated to transpire via pathways such as the activation of NADPH oxidase by immune recognition, which is then followed by the activation of antioxidant systems (e.g., catalase, ascorbate peroxidase, etc.). The objective of the study was to rapidly scavenge H_2_O_2_ and to prevent cellular damage caused by excessive accumulation. A team from Huazhong University of Science and Technology designed an electrochemical sensing microbundle [59], which can be a powerful tool for the in vivo simultaneous investigation of changes in H_2_O_2_, NO, and pH, with a dynamic range up to 0.3 mM, with a low detection limit of 1.24 μM (S/N = 3) and high sensitivity of 9.27 μA/M for H_2_O_2_, which have an increased sensitivity compared to traditional detection methods.

Another team from the University of Castilla-La Mancha has developed an enzyme-free sensor consisting of screen-printed carbon electrodes capable of continuously tracking hydrogen peroxide released by living cells, with not only a good detection limit (24.9 nM), but also an excellent response time (<2 s) [41].

The combination of electrochemical technology and molecular markers provides new ideas for screening disease-resistant varieties. An article from the University of Alberta has reported on the design and development of a biosensor based on anti-*S. sclerotiorum* antibodies as probes immobilized on interdigitated electrodes (IDEs) and sense the binding of the ascospores by label-free non-Faradaic impedimetric detection for sensitive and selective detection and quantifcation of ascospores [42]. Conventional detection methodologies, including bioassay, chromatography, immunoassay, and liquid chromatography-mass spectrometry, are characterised by a number of disadvantages. These include the necessity of costly equipment, the complexity of pretreatment, the stringent requirements for operators, and the extended operation time. Electrochemical sensors are distinguished by their uncomplicated operational processes, rapid response rates, high cost-effectiveness, and high levels of detection sensitivity. These sensors have been demonstrated to expedite the precise and rapid detection of target substances at low concentrations in rapeseed, such as brassinolide, thereby providing a robust means for the monitoring of hormone levels during the growth cycle of rapeseed. Here is Figure 6.

## 7. Technical Bottlenecks in Electrochemical Detection of Plant Diseases

Despite notable progress in electrochemical biosensors, several key technical bottlenecks hinder their large-scale application in plant disease diagnostics, particularly for oilseed crops.

Matrix interference: In the process of plant disease detection, polysaccharides and phenols in plant sap [60,61] are easily adsorbed on the electrode surface, causing signal drift and affecting the accuracy and stability of detection results. This matrix interference problem is a challenge in actual detection, especially in complex samples, how to effectively reduce metabolites interference has become one of the urgent problems to be solved.

Equipment miniaturization bottleneck: Despite the evident potential for portable detectors in field monitoring applications, existing equipment is inadequate in meeting the requirements for long-term monitoring due to its power consumption and stability issues. The high power consumption of these devices invariably leads to their inadequate endurance, whilst their insufficient stability readily triggers data fluctuations and errors, thus limiting their application in actual agricultural production [62,63].

Lack of standardization: The current absence of a unified detection process and quality control standards for plant diseases represents a significant challenge. This issue not only impacts the comparability and reliability of detection results, but also hinders progress towards the industrialisation of related technologies [64]. The application of differing testing methodologies and standards across various research institutes and enterprises has led to a proliferation of testing products within the marketplace. This, in turn, poses a significant challenge to the effective establishment of industrial standards [65].

Lack of highly specific marker molecules: Another major obstacles to field-deployable electrochemical detection of plant pathogens is the absence of robust and highly specific molecular markers [66,67,68]. These markers-typically proteins, peptides, metabolites, or nucleic acid sequences—are essential for differentiating pathogenic species with high confidence in complex plant matrices. However, many fungal pathogens, exhibit high inter-species genomic similarity and intra-species genetic variability, making it difficult to identify unique and conserved biomolecular signatures. For example, genera such as Fusarium, Alternaria, and Sclerotinia share conserved ribosomal DNA and ITS sequences, leading to cross-reactivity in DNA-based assays. Likewise, small molecule metabolites like oxalic acid or galacturonases are secreted by multiple unrelated pathogens, making them insufficient as standalone diagnostic targets. This molecular overlap not only increases the risk of false positives or negatives-especially in asymptomatic or co-infected plant tissues-but also undermines the specificity of biosensors relying on aptamers or antibodies [69]. Moreover, the scarcity of validated, pathogen-specific markers limits the feasibility of multiplex detection systems and hinders the development of high-throughput, automated diagnostic platforms. Without molecular targets that are both pathogen-specific and field-stable, current sensor systems struggle to achieve the precision and scalability required for large-scale agricultural monitoring [70].

## 8. Research Directions and Innovation Pathways

In order to surmount the present limitations and augment the applicability of electrochemical biosensors in the field of plant disease diagnostics, there is a necessity to establish several key research directions and innovation strategies as priorities:

Multidisciplinary cross-fertilization: In the future, optimizing signal parsing algorithms in combination with artificial intelligence (AI) will become an important research direction. For example, the machine learning model developed by a team from Zhejiang University, by establishing an indoor unmanned aerial vehicle (UAV) low-altitude remote sensing simulation platform to obtain thermal, multispectral, and RGB images before and after the artificial inoculation of Mycobacterium smegmatis on oilseed rape leaves, and analyzing the temperature changes through thermal image processing before handing them over to the machine-learning model, the results showed that the classification accuracy of this model to classify the severity of the disease was 90.0% [71]. The application of this technology will greatly improve the efficiency and accuracy of detection, providing strong support for the early diagnosis of plant diseases.

Miniature ex vivo testing equipment: A team of researchers from the University of Alberta has developed an array of microwells activated using nanoelectrodes and another chip for airborne spore detection that can ex vivo detect *S. sclerotiorum* ascospores in the air [72,73]. These devices have the advantages of being lightweight, flexible and non-invasive, and can be used for long-term monitoring without affecting the normal growth of plants, providing a new technical means for early warning and precise prevention and control of plant diseases.

Green detection technology: In the contemporary era, there is a growing awareness of the necessity to protect the natural environment. Consequently, the development and application of green detection technology is set to become a significant trend in the future. In the future, the green development of electrochemical detection technology in oilseed rape disease diagnosis will be centred on the concept of sustainability, and the deep integration of green chemistry and intelligent sensing technology to form a closed-loop detection system that is environmentally friendly and resource-saving. The core direction will focus on the ecological compatibility innovation of electrode materials, the development of new sensing interfaces based on renewable biomass or degradable nanocomposites [49,63,74], the reduction of heavy metal catalyst dependence through biomimetic structural design, and the introduction of optical/biological self-cleaning function to reduce the use of chemical cleaning agents. The greening of the detection process will be reflected in the integration of miniaturized microfluidic chips with low-power sensing systems [46,75], combined with solar-powered or biofuel cell energy supply to achieve zero-emission operation for in situ detection in the field. These technological advances and innovations will not only reduce environmental pollution, but will also be biocompatible and sustainable, in line with the needs of modern agricultural development.

Combined with biological control: Electrochemical detection technology in oilseed rape disease monitoring is being deeply integrated with biological control to promote the establishment of a new model of precise and intelligent green prevention and control. The pathway includes: (1) dynamic monitoring of pathogen signaling molecules and plant defense metabolites using a high-sensitivity electrochemical sensing platform to provide real-time data support for biocontrol; (2) establishment of a quantitative disease-microbe activity model by coupling microfluidic chip and detection unit to intelligently regulate biocontrol fungicide release parameters; and (3) combining sensor arrays and metabolomic data to parse plant-microbe interactions networks and optimize Resistance inducer screening. The breakthrough of nano-electrode technology will promote the synergistic application of enzyme activity detection and fungicide, while the integration of wearable sensing devices and agricultural Internet of Things can build a closed-loop system of “early warning-decision-making-assessment”, realizing the transformation from passive management to active defense [76,77]. Here is Table 2.

## 9. Conclusions

The present paper provides a detailed summary of the technical principles of electrochemical sensors. These principles include the core mechanisms of current-type, impedance-type and potential-type sensors. The paper also includes optimisation of key components such as electrode material selection, biosensor design and signal amplification strategies. By analysing both domestic and international research progress, this paper discusses the typical applications of electrochemical detection technology in the detection of the pathogen of oilseed rape downy mildew, molecular marker analysis of sclerotinia rot, monitoring of metabolites in peanut white mould disease, detection of the pathogen of soybean rust, monitoring of physiological metabolites, and gene marker-assisted breeding.

However, the technological advancement is confronted with various challenges, including matrix interference, limitations in miniaturisation of devices and absence of standardisation during its dissemination. It is therefore recommended that interdisciplinary integration should be a primary focal point for future research efforts. Such integration should encompass the combination of artificial intelligence to optimise signal resolution algorithms, the development of flexible wearable devices for continuous monitoring, and the promotion of green detection technologies to reduce environmental pollution. The interplay between policy and industry has been identified as an area of critical importance. This is predicated on the establishment of standardised detection methodologies, the fostering of integration between agricultural IoT and electrochemical sensors, and the establishment of a national system for the early warning of plant disease.

In essence: the utilisation of electrochemical detection technology holds considerable potential for application in the domain of rapeseed disease detection. Notwithstanding certain technical impediments, through multidisciplinary integration and the development of green technology, electrochemical sensors are anticipated to transition from the laboratory to the field, thereby providing substantial technical support for smart agriculture and sustainable agricultural development, and contributing to global food security and agricultural economic stability.

## Figures and Tables

**Figure 1 foods-14-02881-f001:**
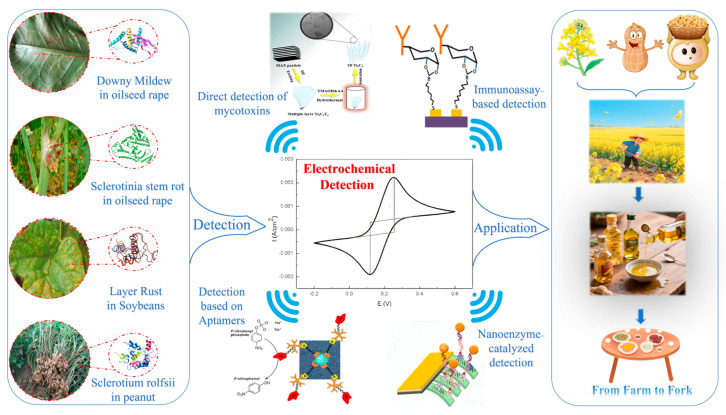
Main hazards of pathogen in oilseed crops and the need for detection.

**Figure 2 foods-14-02881-f002:**
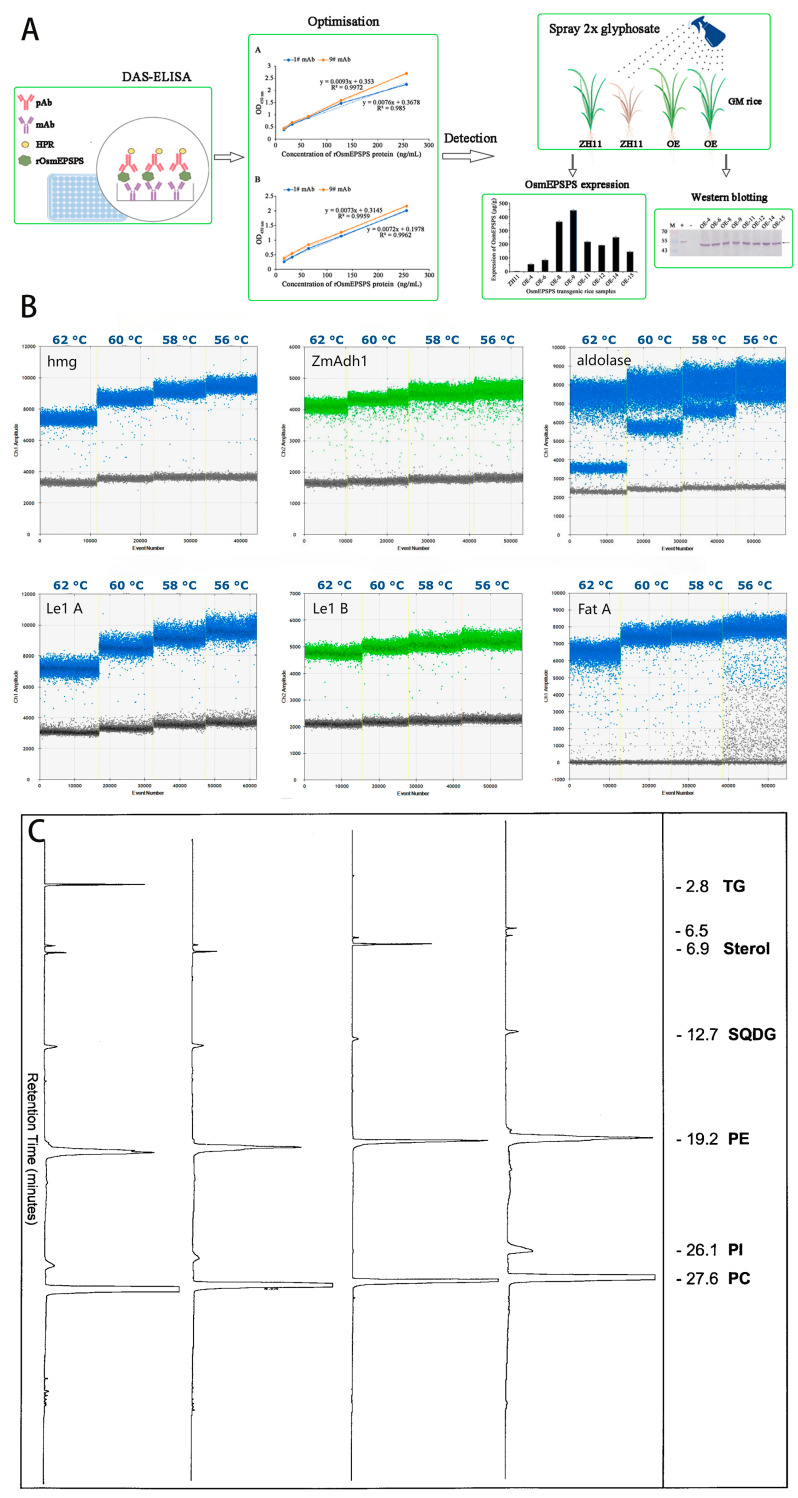
(**A**) OsmEPSPS protein quantification method using double antibody sandwich quantitative ELISA (Ref. [37]. 2025, Luo Biao.); (**B**) emperature gradient ddPCR results for the different taxon-specific assays (Ref. [38]. 2018, Sara Jacchia.); (**C**) HPLC separation and fractionation of the polar lipid extracts derived from different Brassica napus L (Ref. [39]. 2003, Chris-topher Beermann.).

**Figure 3 foods-14-02881-f003:**
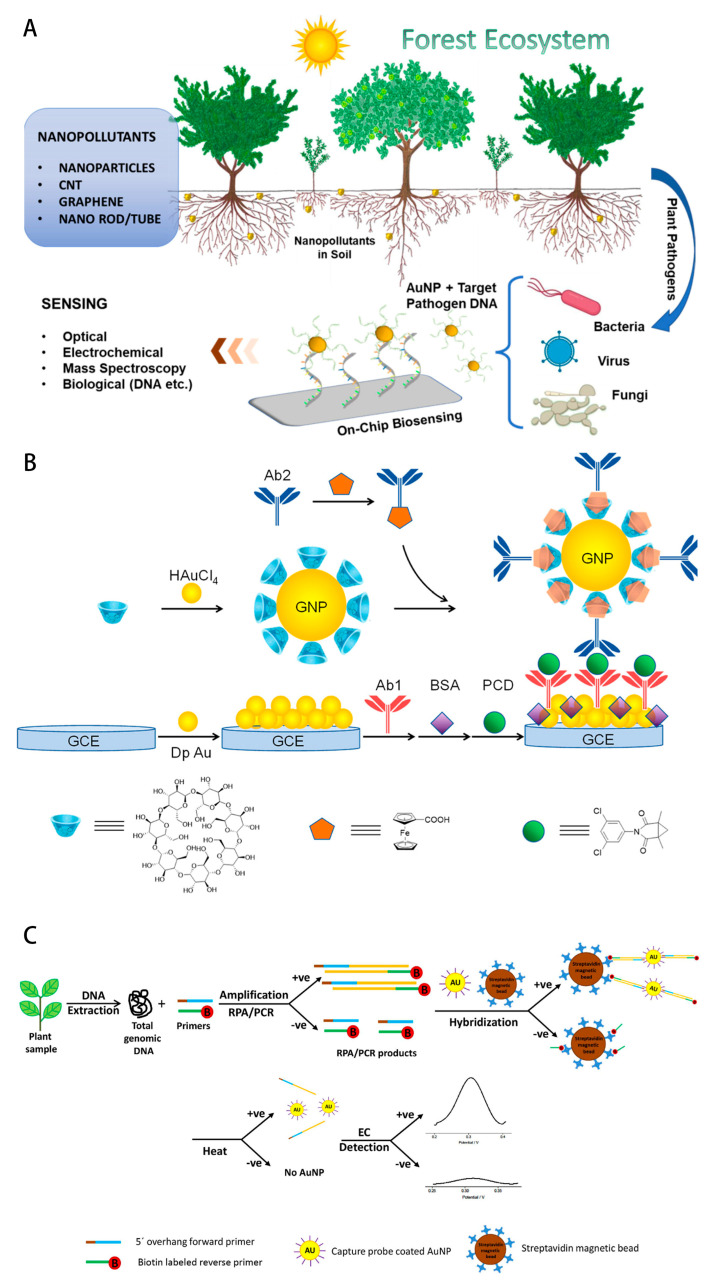
(**A**) SSchematic presentation nanomaterial pollution and use of nanomaterial for plant pathogen detection (Ref. [42]. 2022, Prabir Kumar Kulabhusan.); (**B**) Schematic illustration of the fabrication process of the immunosensor (Ref. [44]. 2025, Qiushuang Ai.); (**C**) Schematic illustration of the electrochemical bioassay for plant pathogen DNA detection (Ref. [43]. 2016, HanYih Lau.).

**Figure 4 foods-14-02881-f004:**
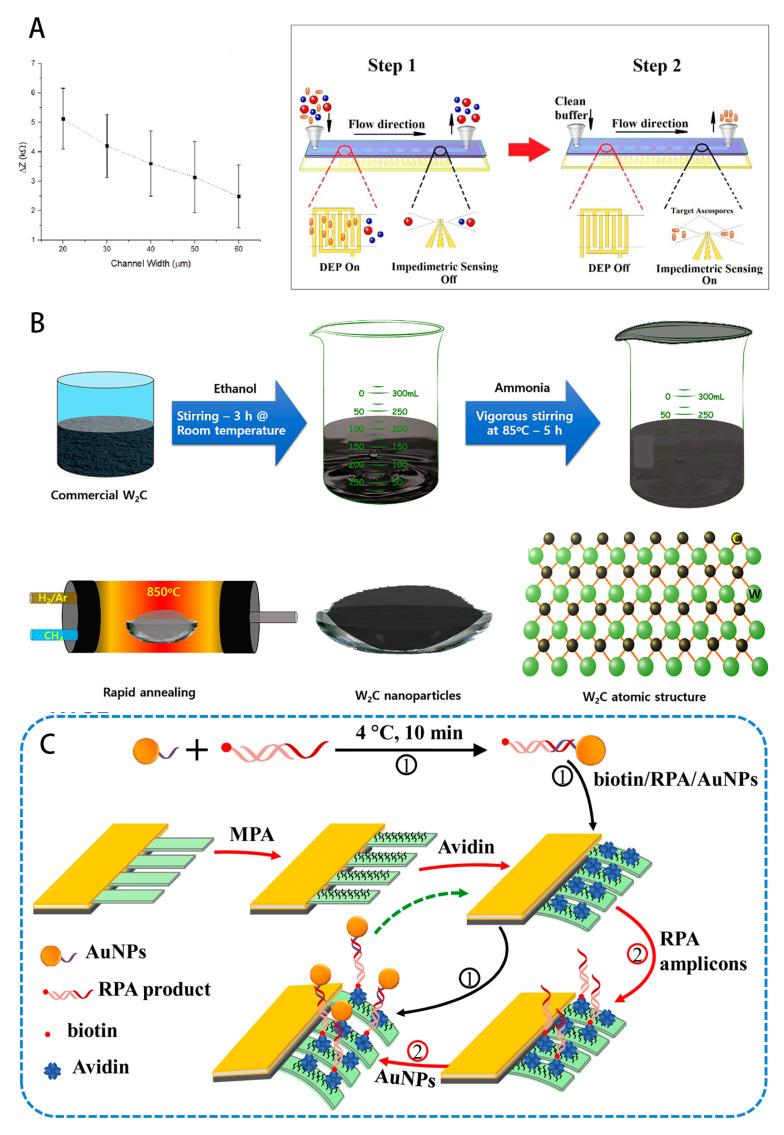
(**A**) Microfluidic microarray device that can be used to measure the impedance change caused by single S. sclerotiorum ascospores (Ref. [51]. 2020, Pedro A.); (**B**) Detection of oxalic acid by molecularly imprinted electrochemical sensor based on tungsten carbide (W2C) nanoparticles (Ref. [50]. 2020, Sajjad Hussain.); (**C**) Dynamic Microcantilever- Recombinase Polymerase Amplification assay Detection of traces of Lactobacillus flavus genomic DNA in oilseed rape genomic DNA (Ref. [47]. 2021, Rong Lei.).

**Figure 5 foods-14-02881-f005:**
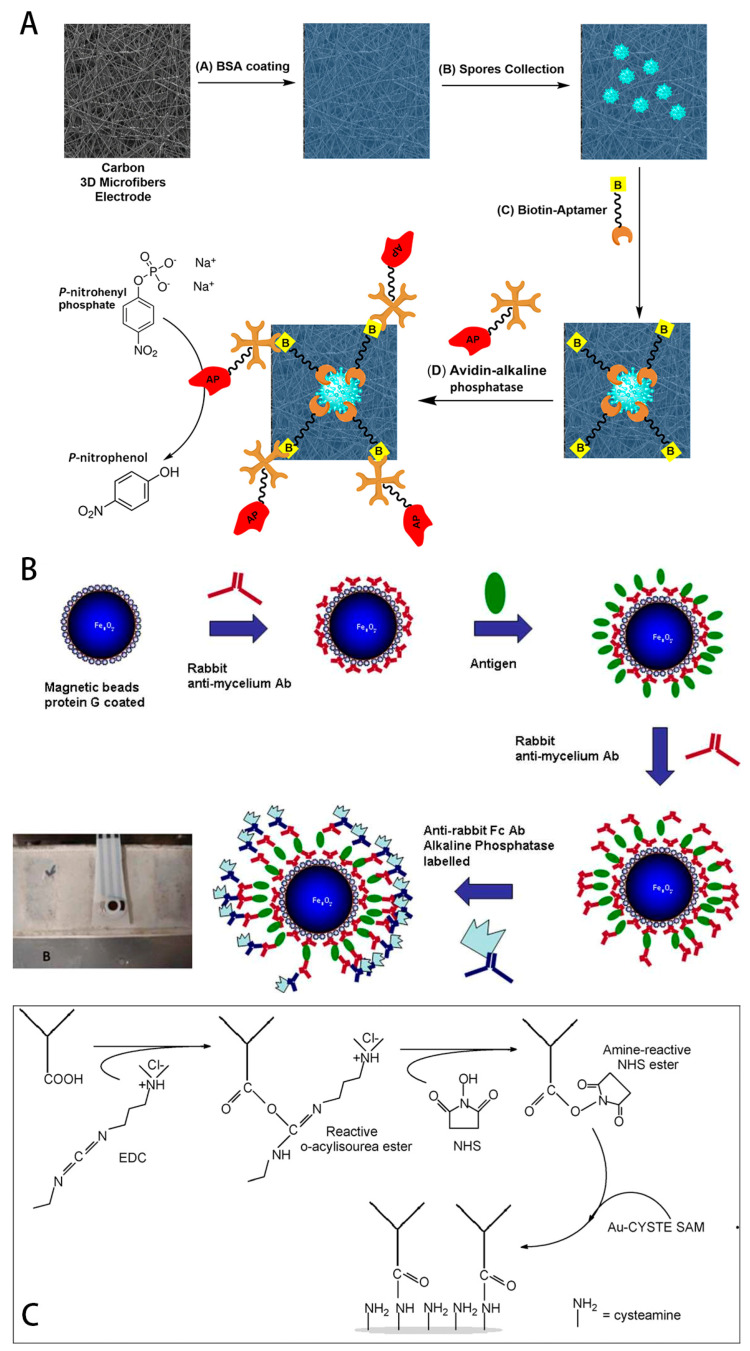
(**A**) Rapid Collection and Aptamer-Based Sensitive Electrochemical Detection of Soybean Rust Fungi Airborne Urediniospores (Ref. [55]. 2021, Vadim Krivitsky.); (**B**) A disposable voltammetric immunosensor based on magnetic beads for early diagnosis of soybean rust (Ref. [53]. 2012, R.K. Mendes.); (**C**) Surface plasmon resonance immunosensor for early diagnosis of Asian rust on soybean leaves (Ref. [54]. 2009, R.K. Mendes.).

**Figure 6 foods-14-02881-f006:**
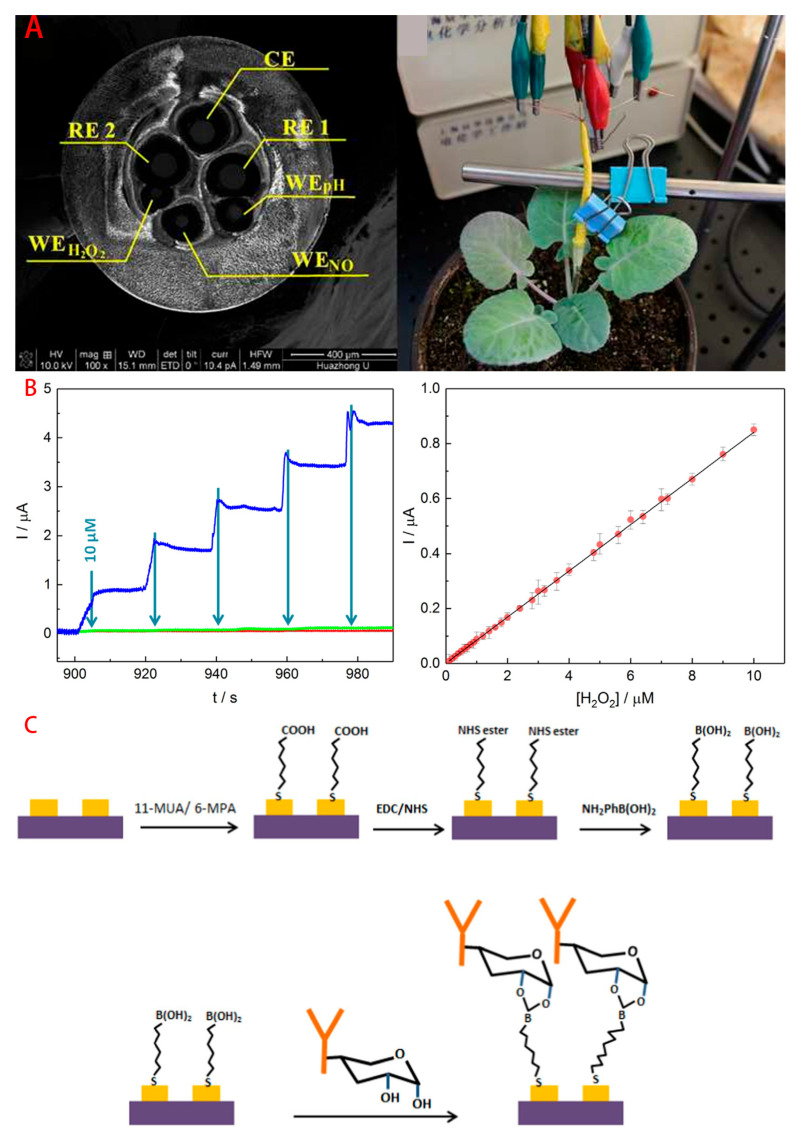
(**A**) Minimally invasive electrochemical microtubule bundles for simultaneous detection of H_2_O_2_,NO and pH changes in plants (Ref. [59]. 2015, Qiong-Qiong Ren.); (**B**) Non-enzymatic screen-printed sensor based on PtNPs@polyazure A for H_2_O_2_ detection (Ref. [41]. 2020, R Jiménez-Pérez.); (**C**) Immuno-impedimetric biosensor forecasting of Sclerotinia Stem Rot of Canola (Ref. [52]. 2018, LianC.T. Shoute.).

**Table 1 foods-14-02881-t001:** Performance comparison between different electrochemical sensors and traditional analytical methods.

Testing Methods	Principle	LOD	Liner Range
Electrochemical sensor	Pt@PAA(DS)/aSPCEs	24.9 nM	0.1–300 mM
Electrochemical sensor	Borosilicate glass@Al/IDE	Single ascospore	≥10 spores/m^3^
Electrochemical sensor	Streptavidin/AuNPs	1 × 10^−5^ ng *L. maculans* DNA	0.315 pg–3.15 ng
Electrochemical sensor	Ab/IDE/SAM	7.8 × 10^4^ spores/mL	0–1.5 × 10^6^ spores/mL
Electrochemical sensor	Au/Ab1/BSA/PCD/Ab2 bioconjugates	1.67 pM	5 pM–0.1 μM
Electrochemical sensor	MIP/W_2_C	0.04 nM	0.1 nM–100 μM
ELISA		20 ng/mL	10^3^–10^6^ spores/mL
Mid-infrared spectroscopy combined with chemometrics		Classification accuracy > 80%	>1504 variables
HPLC-ELSD/GC-FAME		0.025 mg/mL	0.025–1.5 mg/mL

**Table 2 foods-14-02881-t002:** Comparison of research progress and trend analysis.

Research Area	Research Progress	Trends Analysis
Electrode Materials	Graphene, MIPs, gold nanocomposites [47] Nano-enzymes [43], DNA-modified electrodes, biodegradable materials	High sensitivity, low cost, green environment
Biometric element	Aptamers [78,79,80,81], Antibody Modification [42,45] Natural enzymes, DNA probes [44]	High selectivity, fast response
Equipment Development	Portable Tester [75,82], Intelligent Diagnostic System Flexible patch, wireless transmission system	Miniaturized, intelligent, real-time monitoring
Interdisciplinary integration	AI algorithm optimization, microfluidic chips [46,72,82] Machine learning, microfluidic integration	Multi-technology synergy, automated analysis
Green detection technology	Low toxicity reagent research [74] Biodegradable electrodes, natural enzyme systems [76,77]	Sustainable development, environmentally friendly

## Data Availability

No data was used for the research described in the article.

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
