# Peer review of "Electrochemical Biosensors for Oilseed Crops: Nanomaterial-Driven Detection and Smart Agriculture"

_foods, 2025, doi:10.3390/foods14162881_

Round 1

Reviewer 1 Report

Comments and Suggestions for Authors

This review article is well structured and well written. Some comments to be addressed;

  1. Abstract is good. we suggests to add the methodology approach during performing this review: what databases used? keywords? etc.
  2. Introduction is niche, we recommends to highlight the novelty of review along with review scope.
  3. Some yypos existed: (S. sclerotiorum, P. pachyrhizi, S. rolfsii, H. parasitica) --> should be italicized
  4. Please add the methods section: how about article identification, article screening, inclusion and extension criteria
  5. Please give the limitations and implications in your conclusion

Author Response

Dear Editor and reviewers,

Thank you very much for giving us an opportunity to revise our manuscript “Electrochemical Biosensors for oilseed crops: Nanomaterial- Driven Detection and Smart Agriculture”. The number you gave this manuscript earlier was Foods-3768540. We appreciate you and reviewers' time and insightful comments, which have helped us improve the manuscript. Based on the comments we received, careful modifications have been made to the manuscript. All changes were marked in red text. We hope the revised manuscript will meet your magazine’s standard. Below you will find our point-by-point responses to the comments. Furthermore, we made some other changes in the manuscript. These changes will not influence the content and framework of the paper, and here we did not list the changes but marked in red in revised paper.

Reviewer 1:

  1. Abstract is good. we suggest to add the methodology approach during performing this review: what databases used? keywords? etc.

Answer: Thank you for your valuable suggestion. We added a Methodology section after the introduction to include search keywords, article selection criteria, and a description of the comparative database used, such as This review focuses on four major oilseed crop pathogens (S. sclerotiorum, P. pachyrhizi, S. rolfsii, and H. parasitica) as keywords. By comparing disaster-related data and recommended national standards provided by agricultural bureaus in different countries, focuses on……

  1. Introduction is niche, we recommends to highlight the novelty of review along with review scope.

Answer: Thank you for this helpful suggestion. We have revised the introduction section to explain the novelty of this manuscript compared to previous articles and emphasise the scope of this manuscript's review (S. sclerotiorum, P. pachyrhizi, S. rolfsii, and H. parasitica).

  1. Some yypos existed: (S. sclerotiorum, P. pachyrhizi, S. rolfsii, H. parasitica) --> should be italicized.

Answer: Thank you for pointing this out. We have corrected these typographical errors and reviewed the entire manuscript to prevent such problems from recurring.

  1. Please add the methods section: how about article identification, article screening, inclusion and extension criteria.

Answer: Thank you for your valuable feedback. We agree that a clear description of the literature search and selection methodology is crucial for the transparency and reproducibility of our systematic review. We have added a Methodology section (after Introduction section) to this manuscript to describe our literature search and selection strategy.

  1. Please give the limitations and implications in your conclusion.

Answer: Thank you for the suggestion. In the second paragraph of the Conclusion section, we added a brief discussion of future challenges and future work priorities, such as Such integration should encompass the combination of artificial intelligence to optimize signal resolution algorithms.

Reviewer 2 Report

Comments and Suggestions for Authors

Please see in the attachment.

Author Response

(The authors gave the same response as above.)

Round 2

Reviewer 2 Report

Comments and Suggestions for Authors

Please see in the attachment.
